# Laser-induced vapour nanobubbles improve drug diffusion and efficiency in bacterial biofilms

Eline Teirlinck [1,2], Ranhua Xiong[1,2], Toon Brans [1,2], Katrien Forier[1,2,3], Juan Fraire [1,2], Heleen Van Acker [4], Nele Matthijs[4], Riet De Rycke[5,6], Stefaan C. De Smedt[1,2], Tom Coenye [4] & Kevin Braeckmans [1,2,7,8]

Hindered penetration of antibiotics through biofilms is one of the reasons for the alarming increase in bacterial tolerance to antibiotics. Here, we investigate the potential of laser-induced vapour nanobubbles (VNBs) formed around plasmonic nanoparticles to locally disturb biofilm integrity and improve antibiotics diffusion. Our results show that biofilms of both Gram-negative (*Burkholderia multivorans*, *Pseudomonas aeruginosa*) and Gram-positive (*Staphylococcus aureus*) bacteria can be loaded with cationic 70-nm gold nanoparticles and that subsequent laser illumination results in VNB formation inside the biofilms. In all types of biofilms tested, VNB formation leads to substantial local biofilm disruption, increasing tobramycin efficacy up to 1-3 orders of magnitude depending on the organism and treatment conditions. Altogether, our results support the potential of laser-induced VNBs as a new approach to disrupt biofilms of a broad range of organisms, resulting in improved antibiotic diffusion and more effective biofilm eradication.

---

[1] Laboratory of General Biochemistry and Physical Pharmacy, University of Ghent, Ghent 9000, Belgium. [2] Centre for Nano- and Biophotonics, Ghent 9000, Belgium. [3] Laboratory of Toxicology, Ghent University Hospital, Ghent 9000, Belgium. [4] Laboratory of Pharmaceutical Microbiology, University of Ghent, Ghent 9000, Belgium. [5] Department of Biomedical Molecular Biology, VIB Center for Inflammation Research, Ghent University, 9052 Ghent, Belgium. [6] Expertise Centre for Transmission Electron Microscopy, VIB BioImaging Core, Ghent University, Ghent 9052, Belgium. [7] IEMN UMR 8520, Université de Lille, Villeneuve d'Ascq 59652, France. [8] Laboratoire de Physique des Lasers, Atomes et Molécules UMR 8523, Villeneuve d'Ascq 59655, France. Correspondence and requests for materials should be addressed to K.B. (email: Kevin.Braeckmans@UGent.be)

The increasing antibiotic resistance against routinely used antimicrobial agents threatens modern medicine worldwide and has reached alarming levels[1]. One of the reasons for the decreased sensitivity of bacteria towards antibiotics is their capability to form so-called biofilms[2–4]. The increased tolerance of sessile cells is multifactorial and includes avoidance of antibiotic-induced oxidative stress, increased expression of antibiotic efflux pumps, and a reduced penetration rate of antibiotics[5–7]. The latter is sometimes referred to as the biofilm diffusion barrier and has two primary causes. First of all the antibiotic can undergo physicochemical interactions with biofilm constituents, such as polysaccharides, extracellular DNA, or enzymes, causing a decrease in the effective available concentration, and a slower penetration rate[8,9]. The second contribution to the diffusion barrier is of a more physical nature and is a consequence of the fact that sessile cells are packed together inside this biofilm in dense microcolonies of tens to hundreds of micrometers in size[10–12]. Even though antibiotics may be small enough to diffuse between the cells, the net flux towards the deeper cell layers is reduced compared to what is observed in planktonic cultures. It is, therefore, of interest to develop strategies to interfere with the dense biofilm architecture so as to improve the antibiotics flux towards deeper cell layers[13].

A traditional way to interfere with biofilm integrity is by targeting essential components of the biofilm matrix[14]. Examples include dispersin B that was used to degrade poly-*N*-acetylglucosamine (PNAG) in staphylococcal biofilm matrices[15,16] and DNase I to cleave eDNA of *Pseudomonas aeruginosa* and *Staphylococcus aureus* biofilms[17,18]. Another approach is to interfere with the bacterial communication system, called quorum sensing (QS). Many studies reported that QS inhibitors affect the biofilm's structural organization, thereby potentiating antibiotic's efficiency[19–21]. However, a downside of such pharmacological

approaches is their high specificity, i.e., the same treatment cannot be used for a wide range of organisms. This becomes especially problematic in multispecies biofilms in vivo, where one must first elucidate which bacteria and strains are involved in order to select the most appropriate treatment[22]. A second difficulty arises from the fact that many targets for dispersal, such as proteins, are not only present in bacteria but as well in humans, leading to unwanted side effects, such as proteolytic degradation of host-associated proteins[23]. Therefore, it is attractive to think of physical methods to disturb biofilm integrity as they may be more widely applicable, being independent of the biofilm composition. One example is extracorporal shock waves generated by a shock wave generator which were shown to disrupt *P. aeruginosa* and *S. aureus* biofilms, thereby increasing their susceptibility towards ciprofloxacin[24]. Another example is ultrasound therapy, which has been shown to increase gentamicin transport through *P. aeruginosa* biofilms[25]. However, the clinical applicability of unfocused shock waves and ultrasound remains uncertain, as detrimental tissue damage and bleeding can occur[24,26,27]. Therefore, it is of interest to look for more refined physical biofilm disturbance methods with the potential to be applied locally with high spatial control.

Here, we present such a new concept making use of nanotechnology and laser treatment to locally disturb the biofilm integrity, potentiating antibiotics penetration and substantially increasing their effectiveness. The concept is based on laser-induced vapour nanobubbles (VNB) which can subtly but significantly expand the space between sessile cells. As depicted in Fig. 1, biofilms are first treated with a suspension of gold nanoparticles (AuNP) which can gradually penetrate in between sessile cells. When the energy of a high-intensity, short (<10 ns) laser pulse is absorbed by such an AuNP, its temperature can rise to several hundreds of degree, causing the water surrounding the

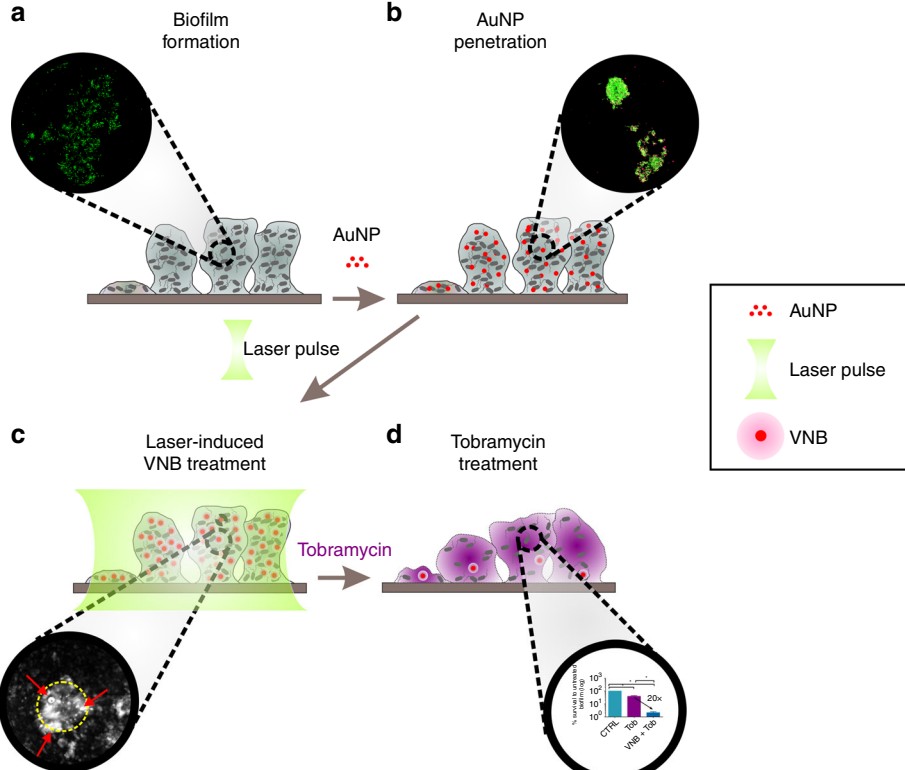

**Fig. 1** Schematic overview of the experimental protocol. **a** Formation of a 24-h bacterial biofilm in vitro on a glass surface. **b** Penetration of AuNP through the biofilm. **c** After absorption of an intense nanosecond laser pulse, vapour nanobubbles emerge around the AuNP. **d** The mechanical force of VNB creates more space between the cells allowing better penetration of antimicrobial agents

particle to quickly evaporate, thus creating a quickly expanding and subsequently imploding water vapour nanobubble[28,29]. The localized shockwaves induced by VNB increase the space between cells, allowing antibiotics to more easily reach the target cells, even deep within the dense cell clusters. A particularly interesting feature of laser-induced VNB is that the heat within the AuNP is efficiently converted into mechanical energy[30]. This means that net transfer of heat into the surrounding healthy tissue is avoided, which is a known problem for the more traditional photothermal therapies[31]. In addition, due to the fine control of laser light and localized action of VNB, the biofilm disturbance effect can be very precisely controlled in space.

We demonstrate the concept with Gram-negative (*Burkholderia multivorans*, *P. aeruginosa*) and Gram-positive (*S. aureus*) biofilms, showing that cationic AuNP can penetrate deep into these biofilms and that laser-induced VNB can substantially disturb their structural integrity. We show that the effectivity of tobramycin added after such a treatment can be increased up to more than 3 orders of magnitude depending on the organism and treatment conditions. In addition, we show in an in vivo *Caenorhabditis elegans* model that there is negligible toxicity of the AuNP at the concentrations used here. Together our findings point to the potential of laser-induced VNB as a new approach to disrupt biofilms of a broad range of organisms, resulting in improved antibiotic diffusion and more effective biofilm eradication.

## Results

**AuNP penetration through biofilms.** The concept of VNB-mediated biofilm disturbance for enhanced penetration of antibiotics was tested on two Gram-negative (*B. multivorans* and *P. aeruginosa*) organisms and one Gram-positive (*S. aureus*). By confocal laser scanning microscopy (CLSM) it was confirmed that

they formed mature biofilms after 24 h with dense bacterial clusters (Supplementary Figure 1). For the generation of laser-induced VNB, we selected cationic AuNP of 70 nm nominal size suitable for VNB generation[29,30]. Dynamic Light Scattering (DLS) size data, zeta potential, TEM image, and UV–VIS spectrum are displayed in Fig. 2. AuNP were added to the mature biofilms at a final concentration of $1.4E + 10$ AuNP mL$^{-1}$ and incubated for 15 min. 3-D confocal images confirmed that these AuNP were indeed able to penetrate into the bacterial cell clusters (Fig. 3 and Supplementary Movies 1,2,3).

**Formation and visualization of VNB inside biofilms.** The next step was to confirm that VNB can be formed in biofilms and investigate their effect on biofilm structure. The optical set-up to generate and detect VNB is schematically shown in Supplementary Figure 2. Biofilms were incubated with AuNP as described above and selected cell clusters were irradiated with a single laser pulse (7 ns, 561 nm) at a laser fluence of 1.69 J cm$^{-2}$. This is above the VNB fluence threshold[32] of AuNP in water (Supplementary Figure 3) to make sure that VNBs are effectively formed. An EMCCD camera was synchronized with the pulsed laser by an electronic signal generator, so that dark field pictures could be taken before, during, and immediately after irradiation.

Dark-field images demonstrated successful VNB formation in all three biofilms incubated with AuNP, as can be seen in Fig. 4 (top 3 rows) where a number of VNB are indicated with red arrows. VNBs are visible as brief (<1 μs typically) localized flashes of light within the laser-irradiated area as they scatter light from the dark field microscope. Most importantly, during VNB formation cell clusters became clearly deformed by the mechanical force exerted by the VNB shock waves. The images after laser irradiation show that the deformation is to a large extent permanent, indicating that the applied force is of sufficient

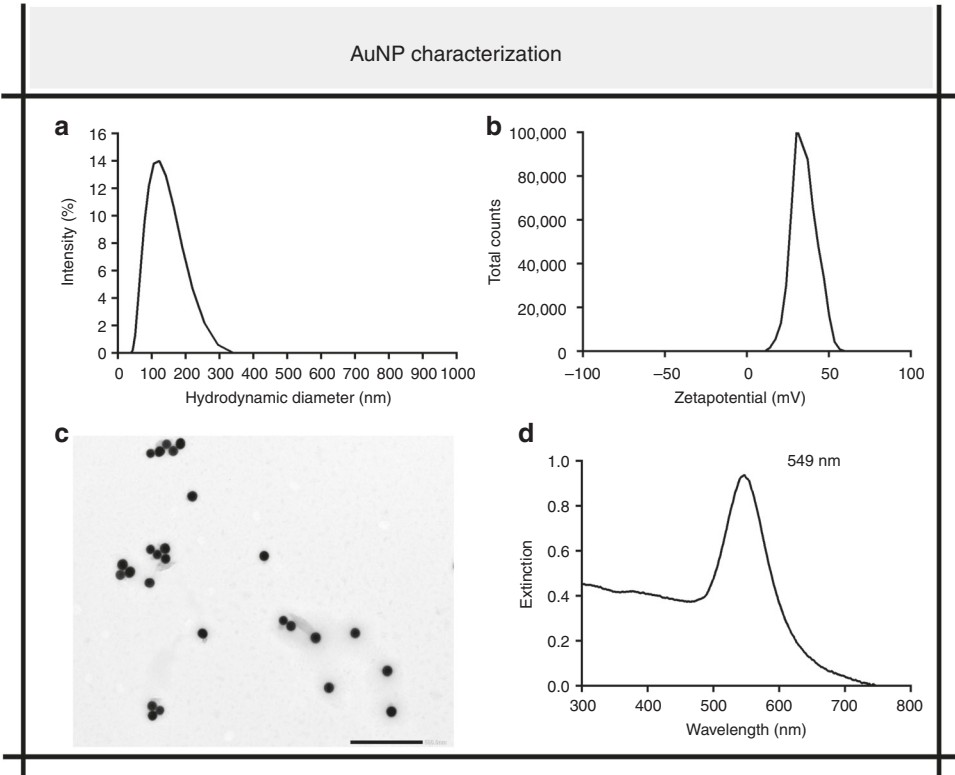

**Fig. 2** AuNP characterization. **a, b** Size distribution (**a**) and zeta potential (**b**) of AuNP, as determined by DLS. **c** Morphological spherical structure of AuNP, visualized by TEM. Scale bar = 500 nm. **d** UV–VIS spectrum of AuNP revealing a localized plasmon resonance peak at 549 nm

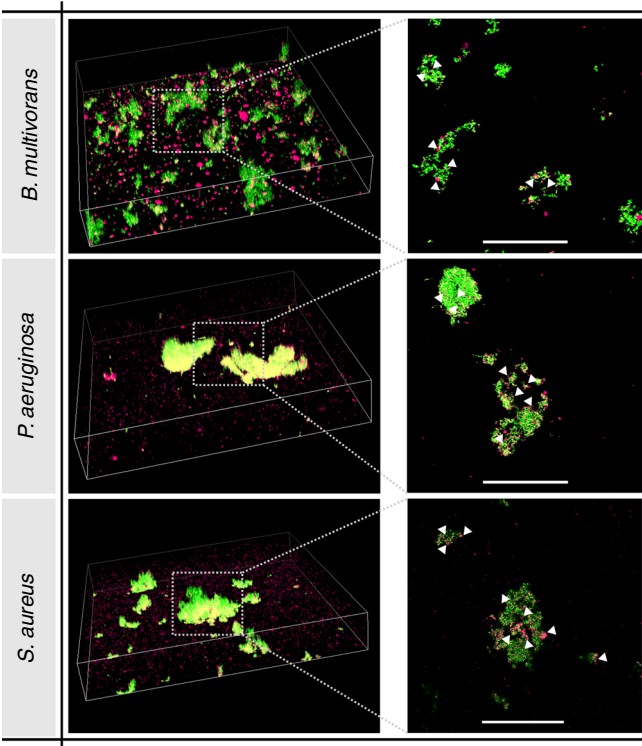

**Fig. 3** AuNP penetration through biofilms. Confocal images of AuNP (magenta) in biofilms of *B. multivorans*, *P. aeruginosa*, and *S. aureus* (green). Left: Large-view 3-D confocal images showing various cell clusters. Width = 212 μm, height = 212 μm, depth = 35 μm (*B. multivorans, P. aeruginosa*), 30 μm (*S. aureus*). Right: Magnified view of the middle plane of selected cell clusters showing the presence of AuNP (some examples are indicated with white arrowheads) throughout the clusters. Scale bar = 50 μm

magnitude to overcome the biofilm's elastic forces. In control biofilms without AuNP, VNB could not be formed upon laser irradiation and consequently the biofilm structure remained unaltered (Fig. 4, bottom row). This clearly demonstrates that it is the combination of AuNP and pulsed laser irradiation that leads to the structural changes in the biofilms.

**Combining VNB-induced cluster disruption and antibiotics.**
Having established that laser-induced VNB can cause substantial deformation of dense cell clusters, we evaluated whether this leads to increased antibiotic-mediated killing of sessile cells. We chose to use tobramycin in concentrations of 32, 16, and 1024 μg mL$^{-1}$ to treat *B. multivorans*, *P. aeruginosa*, and *S. aureus* biofilms, respectively. Note that these are suboptimal concentrations (i.e., they do not result in complete killing of the biofilm, Fig. 5) so that the additional effect of VNB on biofilm killing could be quantified to demonstrate our proof-of-concept. In this case, the biofilms grown in 96-well plates were scanned through the photoporation laser beam so that each location of the sample received 1 laser pulse. Treatment of a single well took ~3 min, which was essentially limited by the 20 Hz laser pulse repetition rate. After laser treatment, tobramycin was immediately added to the biofilms and these were incubated further for 24 h, after which cell survival was quantified by plate counting.

In *B. multivorans* biofilms, pretreatment with VNB significantly increased the effect of tobramycin by approx. 80 times (Fig. 5a). A similar effect was observed in *P. aeruginosa* and *S. aureus* biofilms, with the combined VNB–tobramycin treatment leading to an increased killing efficiency of approx. 20 and 25 times, respectively (Fig. 5b, c). Treatment with AuNP or pulsed

laser irradiation alone did not have a significant effect on the amount of CFUs in any of the biofilms ($p > 0.05$). Neither did the combination of AuNP with tobramycin, or laser treatment with tobramycin, produce any additional effect compared to tobramycin alone. Also VNB treatment by itself did not significantly decrease the number of CFUs. Taken together, these results confirm our hypothesis that the mechanical disruptive effect of laser-induced VNB allows to substantially enhance drug penetration in biofilms, leading to increased killing by at least 1 order of magnitude under the condition of a single laser pulse per location in the biofilm.

It should be noted that the observed effects are not due to heating of the biofilms by the laser-irradiated AuNP as all the thermal energy of the AuNP is converted to mechanical energy of the expanding VNB[33]. This was confirmed by temperature measurements during laser treatment, showing no change in temperature within the sample wells (Supplementary Figure 4).

**The effect of repeated VNB formation.** It is a common observation that AuNP tend to become fragmented upon laser irradiation, so that VNB typically can be formed only once[34–36]. Surprisingly, however, this turned out to be different in biofilms where we noted that VNB could indeed be repeatedly formed upon application of multiple laser pulses. This is illustrated in Supplementary Movies 4 and 5 showing gradual dispersal of cell clusters as VNB are repeatedly formed. While the exact reason for this is unclear, we subsequently evaluated whether forming VNB multiple times could enhance the effect of tobramycin further. The effect of 10× exposure to VNB was evaluated on *P. aeruginosa* and *S. aureus* biofilms (Fig. 6). Tobramycin combined with 10× VNB-treatment, resulted in a more than >3000-fold increased killing in *P. aeruginosa* biofilms when compared to tobramycin alone. This is in contrast to *S. aureus* biofilms, where repeated VNB formation only resulted in approx. 23 times more killing than tobramycin alone. It shows that further cluster dispersal by repeated VNB formation can enhance tobramycin's efficiency further, but that the extent of this is species dependent.

**VNB-treatment enhances transport in biofilms.** In order to investigate our hypothesis that VNB can enhance antibiotic transport through biofilms, the biofilm penetration of FITC-dextrane was compared before and after VNB-formation by confocal microscopy. As displayed in Fig. 7, after irradiating the biofilm with 1 laser pulse, the fluorescence of FITC-dextrane increased towards the center of the bacterial clusters, confirming the hypothesis that VNB-mediated cluster disruption results in increased penetration of molecules into the clusters (see also Supplementary Movies 6-11). It was also observed that repeated VNB treatment could further increase the amount of fluorescence of FITC-dextrane that penetrated in the bacterial clusters.

The penetration of FITC-dextrane into the clusters was quantified off-line using ImageJ (National Institutes of Health). As displayed in Supplementary Figure 5, FITC-dextrane penetration efficiency was determined by comparing FITC-dextrane fluorescence intensity inside versus around the bacterial clusters (after adjusting for the autofluorescence). Image analysis revealed that a single laser pulse increased FITC-dextrane penetration efficiency from 0.48 to 0.81 (and could be increased to 0.84 after 3 laser irradiations). In conclusion, the creation of VNBs inside biofilms can locally disrupt the dense biofilm clusters and cause an increased penetration of molecules into the clusters.

**Bacterial dispersal during VNB-treatment.** As VNB treatment disturbs the integrity of cell clusters, it is of interest to know to which extent bacteria actually become released into the

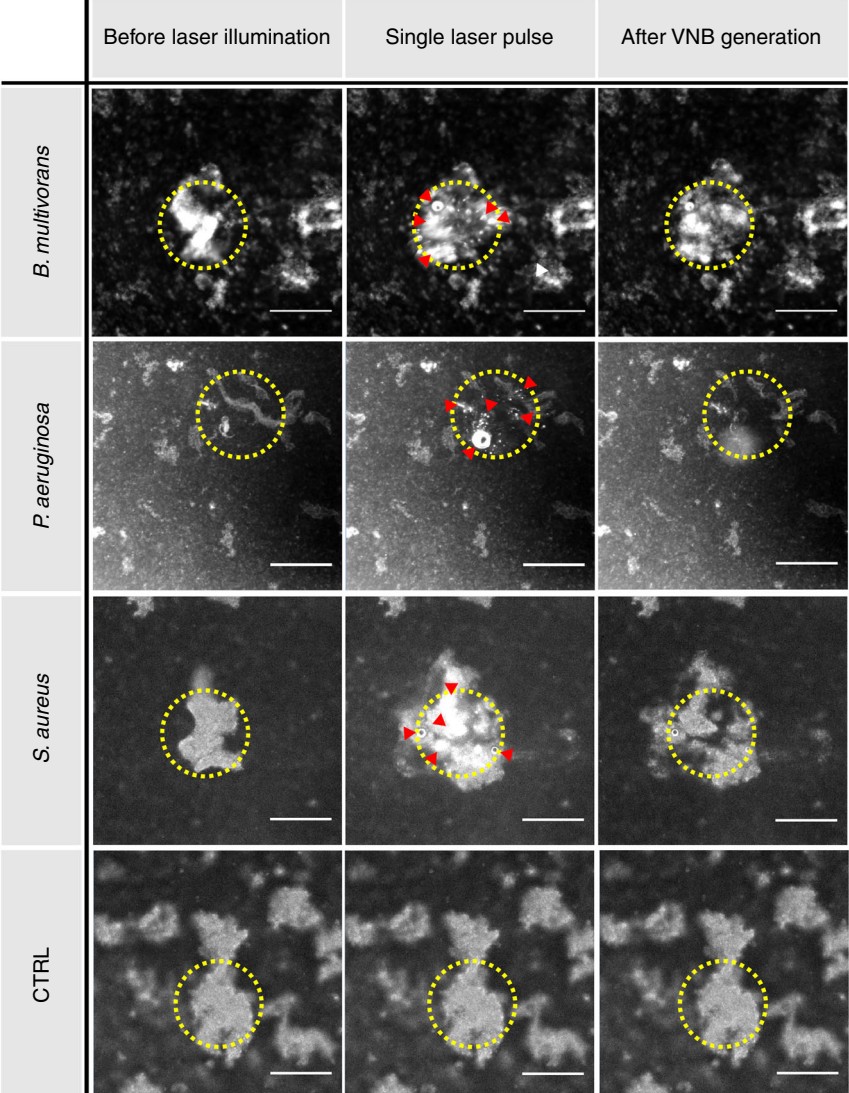

**Fig. 4** Vapour nanobubble mediated biofilm disruption. Dark field microscopy images taken before, during, and about 2 s after a single nanosecond laser pulse of 561 nm laser at a fluence of 1.69 J cm$^{-2}$. The yellow circle indicates the location where the laser pulse is delivered. Some visible VNB are indicated with red arrowheads. CTRL: biofilm cluster without AuNP. Scale bar = 100 μm

supernatant. This is important as dispersed bacteria can disseminate within the host leading to a possible spread of the infection or even septic shock[15]. Furthermore, it has been reported that dispersed cells possess a unique physiology, different from their planktonic and biofilm counterparts, in which they are highly virulent (e.g., against immune system)[37]. To test this, the number of colony forming units in the supernatants and biofilms of both *P. aeruginosa* and *S. aureus* was determined after 1× and 10× VNB treatment via plate counting. No significant differences were found between the numbers of CFUs in the supernatant with or without VNB treatment (Fig. 8). This shows that VNB treatment causes localized disturbance of the biofilm cell clusters, without spreading of bacteria into the environment.

**In vivo toxicity assessment in *Caenorhabditis elegans*.** Given the positive results of VNB treatment, and with an eye on potential in vivo application to biofilm-related infections, we finally tested the toxicity of AuNP at the concentration used here in an in vivo *C. elegans* model. Both pristine AuNP as well as laser-irradiated AuNP were evaluated. A synchronized population of L4 stage *C. elegans* nematodes was fed with either AuNP (final concentration of 1.4E + 10 AuNP mL$^{-1}$) or OGM medium and incubated for 3 days. The amount of living/dead nematodes was determined by light microscopy every 24 h. No significant differences were found in survival after 24, 48, and 72 h in *C. elegans* relative to the control (Fig. 9). It gives the first indication that the (laser-irradiated) AuNP do not cause acute toxic effects in the concentration used in the present study.

## Discussion

CLSM images revealed that cationic 70 nm AuNP could efficiently penetrate deep into the dense cell clusters of *B. multivorans*, *P. aeruginosa*, and *S. aureus* biofilms. Many studies state that biofilms impede NP penetration because of their multicomponent matrix and the presence of dense cell clusters[38,39]. However, at the same time, efficient penetration of NP through biofilms has also been reported[40]. Indeed, NP diffusion through biofilms is highly dependent on physicochemical characteristics such as NP surface charge and size[41]. For example, Duncan et al. reported that cationic charges of amine-functionalized silica present on the outer layer of nanocapsules, enabled them to penetrate *Escherichia coli* biofilms and deliver their therapeutic

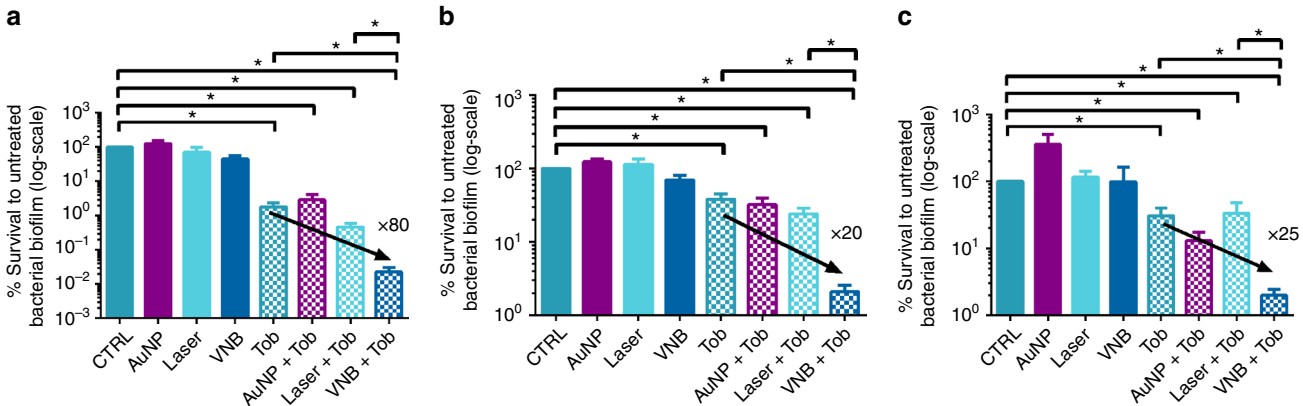

**Fig. 5** Anti-biofilm effect of combined treatment of 1× VNB formation and tobramycin. Data shown are log of the % survival CFU/biofilm compared to the untreated control (average ± SEM) in **a** *B. multivorans*, **b** *P. aeruginosa*, and **c** *S. aureus* biofilms. Control: 0.9% NaCl solution, AuNP: only addition of 70 nm spherical gold nanoparticles, Laser: only laser treatment (each location of the biofilm received 1 laser pulse), VNB: addition of AuNP with subsequent laser treatment created vapour nanobubbles, Tob: tobramycin treatment for 24 h at 37 °C, AuNP + Tob: addition of AuNP with subsequent tobramycin treatment, Laser + Tob: laser and subsequent tobramycin treatment, VNB + Tob: addition of AuNP with subsequent laser and tobramycin treatment. CTRL contained $9 \times 10^6 \pm 3 \times 10^6$ CFU/biofilm, $2 \times 10^7 \pm 1 \times 10^7$ CFU/biofilm, or $2 \times 10^6 \pm 2 \times 10^6$ CFU/biofilm for *B. multivorans*, *P. aeruginosa*, or *S. aureus* biofilms respectively. Each antibiofilm effect was tested in 6 biological repeats and each biological repeat consisted of 3 technical repeats ($n = 6 \times 3$) ($p < 0.05$; The Shapiro–Wilk test was used to test the normality of the data sets. The one-way analysis of variance test and independent samples *T*-test were used for normal distributed data. The Kruskal–Wallis test and Mann–Whitney *U* test were used for non-normally distributed data)

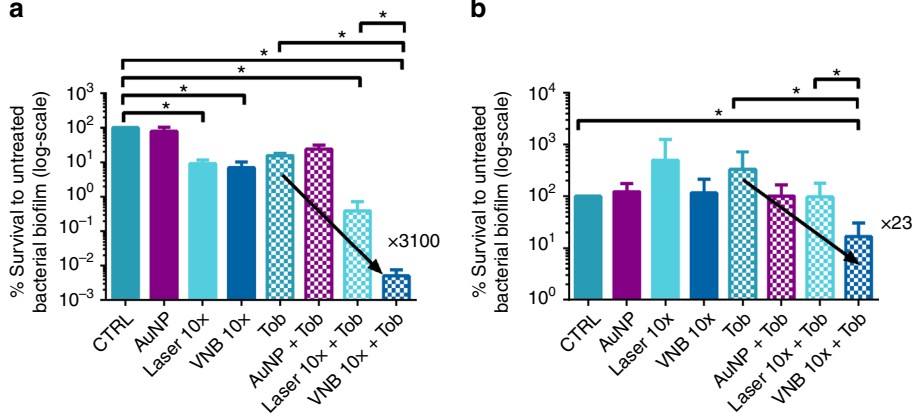

**Fig. 6** Biofilm survival after 10× VNB formation followed by tobramycin treatment. Log of the % survival CFU/biofilm compared to the untreated control (average ± SEM) in **a** *P. aeruginosa* and **b** *S. aureus* biofilms. Control: 0.9% NaCl solution, AuNP: only addition of 70 nm spherical gold nanoparticles, Laser: only laser treatment (each location of the biofilm received 10 laser pulses), VNB: addition of AuNP with subsequent laser treatment to form vapour nanobubbles, Tob: tobramycin treatment for 24 h at 37 °C, AuNP + Tob: addition of AuNP with subsequent tobramycin treatment, Laser + Tob: laser and subsequent tobramycin treatment, VNB + Tob: addition of AuNP with subsequent laser and tobramycin treatment. CTRL contained $2 \times 10^7 \pm 7 \times 10^6$ CFU/biofilm or $1 \times 10^6 \pm 3 \times 10^5$ CFU/biofilm for *P. aeruginosa* or *S. aureus* biofilms, respectively. Each antibiofilm effect was tested in 4 biological repeats and each biological repeat consisted of 3 technical repeats ($n = 4 \times 3$) ($p < 0.05$; The Shapiro–Wilk test was used to test the normality of the data sets. The one-way analysis of variance test and independent samples *T*-test were used for normal distributed data. The Kruskal–Wallis test and Mann–Whitney *U* test were used for non-normally distributed data)

antimicrobial payload[42]. Li et al. showed that surface charge dictated the location of differently charged quantum dots inside biofilms, with cationic QDs being completely dispersed throughout the biofilm, whereas negatively QDs were bound at the outer biofilm layers[43]. Giri et al. found that high penetration through *S. aureus* and *P. aeruginosa* biofilms could be obtained by decorating the AuNP surface with positive ligands[44]. Peulen and Wilkinson examined the importance of NP size on biofilm penetration and found that biofilm mobility is diminished in function of increasing NP radius[41]. In addition, Forier et al. were able to determine the NP size cut-off for optimal penetration into dense biofilm clusters of *B. multivorans* and *P. aeruginosa*, and concluded that PEGylated NPs up to 0.1–0.2 μm are able to

penetrate these biofilms to the same extent as small molecules[45]. Generally these findings are in agreement with the present study, where efficient penetration of positively charged AuNP of 70 nm was observed in all three biofilms. Yet it remains of fundamental interest to evaluate in future work the biofilm penetration potential of AuNP with different sizes and surface functionalizations, including PEGylated ones, to see which particle properties are best for VNB-mediated cluster disruption[46].

By incubating the biofilms with AuNP and applying subsequent pulsed laser irradiation, it was possible to create VNB inside all three studied biofilms. VNBs are generated by illuminating AuNP in biological tissues with pulsed laser light, leading to evaporation of the surrounding water into VNB[28]. Considering

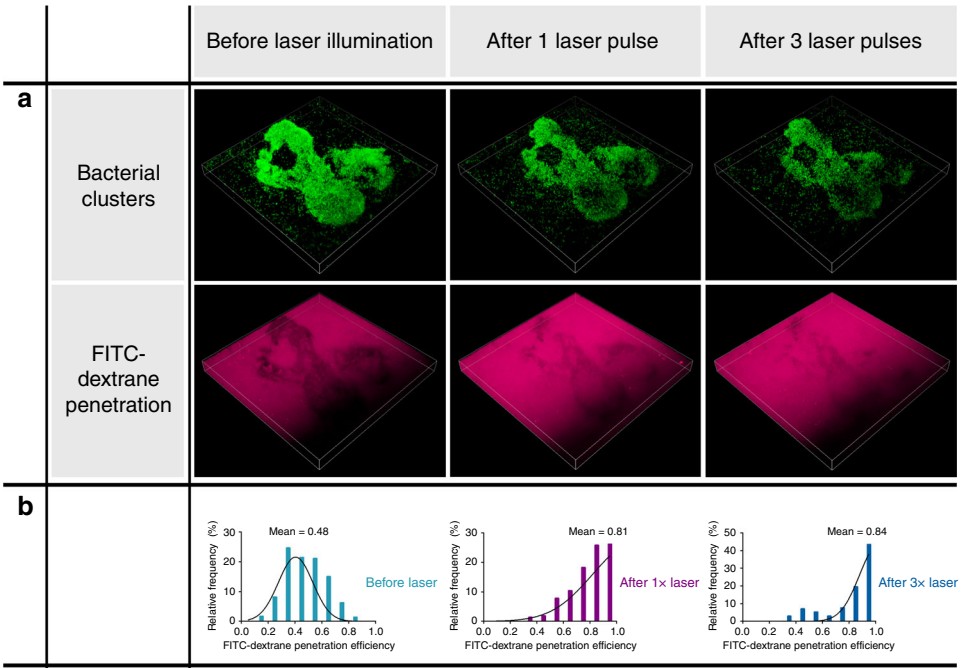

**Fig. 7** VNB-formation enhances the penetration of molecules into biofilms. **a** FITC-dextrane 4 kDa penetration in exemplary bacterial cell clusters of *P. aeruginosa* biofilms before and after VNB-treatment. Bacteria are displayed in green, while FITC-dextrane is depicted in magenta. Width = 212 μm, height = 212 μm, depth = 18 μm. **b** 3-D analysis of FITC-dextrane penetration into a total of 20 clusters (*n* = 5 × 4) was quantified according to Eq. (1). The histograms show that FITC-dextrane penetration was markedly enhanced following 1× or 3× laser treatment

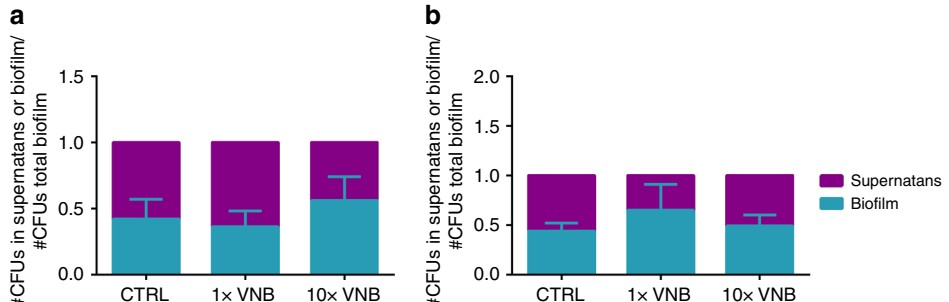

**Fig. 8** Bacterial dispersal during VNB-treatment. The number of cells in the supernatants of **a** *P. aeruginosa* and **b** *S. aureus* biofilms was measured after 1× and 10× VNB treatment by plate counting (average ± SEM). No significant differences were found between any of the tested conditions (*p* > 0.05). Each condition was tested in 3 biological repeats and each biological repeat consisted of 3 technical repeats (*n* = 3 × 3). Statistical analysis was performed with the Shapiro–Wilk test and Kruskal–Wallis test

the high water content of biofilms and the fact that biofilms can be loaded with the aforementioned AuNP, VNB could be successfully formed.

Applying 1× VNB treatment already interfered noticeably with the biofilm structure. This is due to the mechanical impact of the laser-generated VNB which rapidly expand and implode, causing high-pressure shock waves[28]. As shown in Fig. 4, besides the formation of different nanobubbles, also micron-sized bubbles were observed, which may be responsible for the more extensive disruption of cell clusters (compared to the more localized disorganization induced by VNB). Formation of the bigger sized bubbles is likely caused by aggregated AuNP (biofilm-induced AuNP aggregation), while the nanobubbles originated from individual AuNP. Efficient penetration of AuNP towards the center of biofilm clusters is believed to be a keystone of the treatment's success, because in this way the VNBs are created inside the clusters, enabling a disruption force from within—

instead of only disrupting the biofilm from the outer edges. It is important to note that this technique is different from the more traditional photothermal killing of bacteria, where the combined use of AuNP and laser irradiation kills bacteria by producing heat[31,47,48]. Despite numerous in vitro studies reporting the effective eradication of biofilms by photothermal treatment[49–54], translation into clinical practice may be difficult because of heat diffusion into the surrounding healthy tissue[31,51]. One study reported on the destruction of *Bacillus subtilis* biofilms grown on a gold-coated substrate following pulsed laser irradiation. Laser irradiation induced localized melting of the gold layer, resulting in the formation of a fast liquid jet and fragmentation of the gold layer[49]. While this affected the biofilm structure, such an approach is limited to applications where biofilms are formed on gold-coated surfaces. The approach presented here, however, is more versatile in the sense that AuNP can be added to any biofilm irrespective of the surface on which it is formed and even to

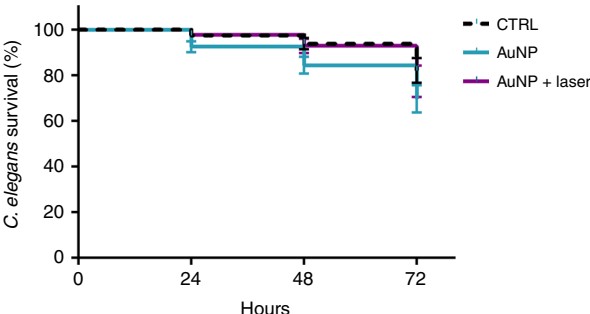

**Fig. 9** In vivo toxicity assessment in *Caenorhabditis elegans*. Kaplan–Meier survival plot of *Caenorhabditis elegans* in the presence of OGM medium (CTRL), pristine 70 nm AuNP (AuNP), or laser-irradiated AuNP (AuNP + laser). Survival was assessed after 24, 48, and 72 h exposure (average ± SEM). No significant differences were found between the different groups. Each condition was tested in 4 biological repeats and each biological repeat consisted of 6 technical repeats ($n = 4 \times 6$). Statistical analysis was performed with the Shapiro–Wilk, Kruskal–Wallis, and ANOVA tests

non-surface attached biofilm aggregates. As such it could be applied to e.g., biofilm infections of wounds or dental root canals. A particular advantage of VNB to sensitize biofilms is that no heat transfer into the surrounding tissue occurs because the heat is efficiently converted into mechanical energy of the expanding VNB[30]. In addition, due to the fine control of laser light and localized action of VNB, this concept allows very precise localized treatment thus minimizing potential harm to the surrounding healthy tissue. One might even consider the use of bacteria-targeted Au to enhance the specificity even further[31,47,50].

Other techniques such as shock waves and ultrasound have also been investigated to render biofilms more sensitive to antibiotics. In the study of Gnanadhas et al., the use of shock waves increased *P. aeruginosa* and *S. aureus* biofilm sensitivity to ciprofloxacin by 100 to >1000 times[24]. In the study of Kopel et al., the combination of surface acoustic waves and gentamicin could reduce the number of CFU of 48 h-old *P. aeruginosa* biofilms by 2 log compared to the effect of gentamicin alone[27]. Some successes have been achieved using low frequency ultrasound as well, as summarized in a recent review[55]. However, tissue damage and bleeding associated with the use of these approaches are of particular concern and because of this, treatments that are effective in vitro can currently not be applied in vivo[24,26,27]. Clearly, much work still needs to be done to further validate our new concept of VNB-mediated biofilm disruption as well, including performing in vivo studies. Considering the inherent limited penetration of light into tissue, the most likely applications are the treatment of topical infections, such as the treatment of wound infections. In this application, the AuNP will likely be applied to the wounds as a topical suspension. After some incubation time, this can be followed by a gentle rinsing step with a washing solution such as physiological saline to remove excess unbound AuNP. In the next step, the infected area will be irradiated with pulsed laser light to generate VNB. Due to the unsurpassed fine control of laser light, we expect that unwanted side effects like tissue damage can be reduced to a minimum. The use of light comes at its expense as well, which is that light penetration into tissues is fairly limited especially in the visible range. Although we used green-yellow light in this study (561 nm), it is important to note that this is not inherent to VNB generation, but only depends on the sensitizing particle being used. For instance, by switching e.g., to gold nanorods, the absorbance spectrum shifts to NIR wavelengths, allowing deeper penetration into tissues. While treatment of wounds with pulsed laser light is currently not performed to our knowledge, it is of note that pulsed lasers that are used for tattoo

removal or port-wine stain treatment[56,57] have the right characteristics for VNB generation, so that at least from a technology point the procedure should be clinically feasible. In the last step, the required antimicrobial agent will be administered to the VNB-treated wounds in the same way as is normally done in wound care. It is of note that for the treatment's success, it will be crucial that the entire infected region is treated with AuNP and tobramycin and irradiated by pulsed laser light in order to reduce the infection. By scanning the laser beam in a (somewhat overlapping) raster pattern, a complete coverage of the infected area can be assured. In addition, one could scan the wound multiple times (as it was observed that repeated VNB-formations inside biofilms is possible) in order to minimize the chance that some small parts would be missed, e.g., due to slight movement by the patient during the scanning procedure. Another potential application is the treatment of dental root canal infections, for which biofilm removal by laser-induced cavitation of the rinsing fluid is already being applied in the clinic[58]. Still, complete biofilm removal remains challenging and often incomplete due to the fact that fluid cavitation only affects the biofilm from the outside. Instead, if combined with AuNP to induce VNB from within the biofilms, it is expected that biofilm removal would become more effective. The AuNP can be dispersed in the disinfectant solution that is normally used during the laser procedure so that no extra steps will be needed. This should lead to more efficient eradication of remaining biofilm cells not only because of VNB shockwaves, but also due to a better penetration of the disinfectants that are present during the procedure.

For all types of biofilms studied, the combined use of VNB and tobramycin has a greater effect than tobramycin alone. Indeed, the use of VNB significantly increased the effect of tobramycin by ~80, ~20, and ~25 times in case of *B. multivorans*, *P. aeruginosa*, and *S. aureus* biofilms, respectively. This is in line with our hypothesis that VNB treatment increases the drug flux towards deeper cell layers by creating more space between the cells. Indeed, it was observed that VNB cause an influx of molecules into the clusters after laser irradiation. Repeated VNB treatment resulted in an even bigger flux of molecules towards the center of the clusters. In addition, it may be that increased diffusion of oxygen contributed to tobramycin's improved efficiency as well. Indeed, it has been shown that biofilms also impede oxygen diffusion, leading to different oxygenation levels throughout the biofilm, which play a part in the biofilm tolerance towards antimicrobials such as tobramycin[6]. Besides increasing net mass transport, disruption of biofilms could also have an impact on other tolerance mechanisms that are derived from their complex three-dimensional structure[8]. In future work, the effect of VNB-disruption on persisters will be studied. This dormant, multidrug-tolerant subpopulation of biofilm cells have been proposed as an important factor in biofilm resilience[59]. It would be interesting to see whether the increased transport of antibiotics by VNB-mediated disruption of biofilms would affect the number of persisters present in the biofilm population and their ability to sustain antibiotic killing. In order to better understand the underlying mechanisms of VNB-mediated biofilm disruption, future work will also assess the viscoelastic change of biofilms after VNB-treatment. Here, macroscopic rheology measurements could be combined with Single Particle Tracking microrheology experiments to come to an in-depth understanding of the viscoelastic changes by laser-induced VNB[60,61].

It is to be noted that we chose in this study to administer tobramycin after laser treatment to avoid any detrimental effect of the laser light on tobramycin. Clearly, in future work, it is of interest to evaluate if the treatment's efficiency could be further enhanced if the antimicrobial agent is present during VNB-formation, thereby having immediate access to the ruptured cell clusters.

During the experiments we rather surprisingly found that VNB could be formed multiple times in biofilms upon repeated laser irradiation from the same batch of AuNP. This was unexpected as we and others have found that nanosecond laser pulses induce fragmentation of AuNP[62], likely due to melting of AuNP and subsequent surface evaporation into smaller sized fragments[63]. The net result is that VNB can typically be formed only once from AuNP. The reason why VNB can be formed multiple times in biofilms remains unclear at this point, but it may be that fragments are still formed but held closely together due to the matrix components. Indeed, it has been previously shown that agglomerates of smaller AuNP can form VNB similar to a single larger AuNP[64]. In addition, it may be that new AuNP become available to laser irradiation as the clusters gradually disperse upon application of multiple laser pulses. Whatever the underlying reason, by forming VNB multiple times, the cluster integrity could be further compromised. In combination with 10× VNB formation, tobramycin efficiency could be enhanced >3000 fold in *P. aeruginosa* biofilms. In case of *S. aureus* biofilms, however, repeated VNB treatment did not further increase the efficiency of tobramycin compared to 1× VNB treatment. It may point to the fact that for *S. aureus* biofilms, probably other mechanisms, besides hindered biofilm diffusion, contribute to biofilm resistance, such as enzymatic alteration of tobramycin[65]. If this turns out to be the case, the use of nanocarriers, such as liposomes, that shield the encapsulated antibiotic from degradation and/or interactions with the biofilm matrix could be a potentially useful strategy.

Although substantial cluster disruption was noticed when biofilms were treated with 10× VNB, no dispersal of bacteria in the supernatant occurred. The aim of classic dispersal agents, such as DNase I, is to disturb tight biofilm architectures and induce concomitant release of bacteria, in order to increase their sensitivity towards antibiotics. Nonetheless, one major drawback of these methods is the possible re-colonization of the dispersed bacteria, which could lead to life-threatening sepsis or biofilm formation in other locations of the human body[15,23]. VNB treatment is highly controlled and causes only local biofilm deformation, without spreading of bacteria in the environment, and hence poses less risk of spreading the infection and causing disease exacerbation.

Our data showed that adding AuNP alone did not cause a significant decrease of cell viability in any of the studied biofilms. While it points to the biocompatible nature of AuNP, we additionally assessed AuNP toxicity in an in vivo *C. elegans* model. The good correlation of the invertebrates endpoints to rat LD50s[66] and their highly conserved toxicity pathways with humans[67], make them an excellent model for in vivo whole organism toxicity testing[68–70]. Furthermore, it has been shown that *C. elegans* is a well-suited platform to study the in vivo toxic effects of various types of AuNP[71,72]. We showed that no significant differences were found in the nematode survival between control and AuNP treated groups, highlighting the non-toxic nature of this approach. Also laser-irradiated AuNP, which are fragmented into smaller pieces, did not cause any noticeable toxicity. Spurred by these positive results, future research should also assess the AuNP toxicity in humans in clinical trials, depending on the intended application, in order to cover human-specific toxicological pathways. Currently, different Au-based particles have entered clinical trials, such as Phase-II Aurimmune (CytImmune®) and Aurolase (Nanospectra Biosciences®) for the treatment of solid tumors and silica-gold nanoparticles for the treatment of plaques[73–75].

In conclusion, increasing antibiotic penetration through biofilms via laser-irradiated AuNP is a very promising route to combat the problem of biofilm-related infections. Depending on the type of biofilm and the conditions used, the effect of tobramycin could be enhanced over 3 orders of magnitude. To fully unravel the potential of this new concept, it will be of interest in future research to explore its potential for relevant in vivo applications that are compatible with laser treatment, such as wound- and root canal-related biofilms infections. Furthermore, because we have shown that VNB-treatment is a robust technique which can render both Gram-negative and Gram-positive biofilms more sensitive to tobramycin, it is of interest to test the concept in multispecies biofilms.

## Methods

**Materials and strains**. *B. multivorans* LMG18825 and *P. aeruginosa* LESB58 (LMG 27622) were obtained from the BCCM/LMG bacteria collection (Ghent University, Belgium), while *S. aureus* Mu50 was obtained from ATCC (Manassas, VA). Mueller Hinton Agar/Broth, Lysogeny Agar/Broth and pure Agar were purchased from Lab M Limited (Lancashire, UK). Simulated wound fluid (1:1 fetal bovine serum: 0.9% NaCl (w/v) in 0.1% Peptone) was prepared in-house. Fetal bovine serum was purchased from HyClone™ (Pierce, Rockford, IL, USA), NaCl was obtained from Applichem (Darmstadt, Germany) and Peptone was derived from BD Diagnostics (New Jersey, USA). Nematode Growth Medium (NGM) agar plates (3 mg mL$^{-1}$ NaCl, 17 mg mL$^{-1}$ Agar, 2.5 mg mL$^{-1}$ Bacto Peptone, 5 mg mL$^{-1}$ cholesterol, 134 mg mL$^{-1}$ KPO$_4$ buffer pH 6, 120 mg mL$^{-1}$ MgSO$_4$, 110 mg mL$^{-1}$ CaCl$_2$) and M9 buffer (3 g L$^{-1}$ KH$_2$PO$_4$, 6 g L$^{-1}$ Na$_2$HPO$_4$, 0.5 g L$^{-1}$ NaCl, 1 g L$^{-1}$ NH$_4$Cl) were prepared in-house. Cholesterol, potassium phosphate buffer pH 6 (108.3 g L$^{-1}$ KH$_2$PO$_4$, 35.6 g L$^{-1}$ K$_2$HPO$_4$), MgSO$_4$, CaCl$_2$, KH$_2$PO$_4$, Na$_2$HPO$_4$ were purchased from Sigma-Aldrich (Saint Louis, MO, USA). NH$_4$Cl was obtained from Fisher Scientific (Loughborough, Leics, UK). AuNP (Nanopartz, USA) were diluted 1:60 in ultrapure water prior to use. According to the manufacturer's information, AuNP were conjugated with a myriad of different covalent SH-branched and straight amine polymers with different MW from 5 to 40 kDa, all manufactured by Nanopartz. As purification, a multistep dialysis was performed resulting in less than 0.01% w-w reactants. SYTO59 was obtained from Thermo-Fisher Scientific (Eugene, USA).

**Biofilm formation**. Biofilms were grown aerobically in 96-well SensoPlates™ (Greiner Bio-One, USA) with microscopy grade borosilicate glass bottom for 24 h at 37 °C. Briefly, biofilms of *B. multivorans*, *P. aeruginosa*, or *S. aureus* were grown in Mueller Hinton Broth, Lysogeny Broth, or simulated wound fluid, respectively. The wells of the 96-well SensoPlate were filled with 100 μL of the bacterial suspension, and incubated at 37 °C. After 4 h, the adhered cells were washed with physiological saline (0.9% NaCl (w/v)), covered with medium and incubated for another 20 h at 37 °C.

**Evaluation of the biofilm structure by CLSM**. To confirm the presence of a biofilm, laser scanning confocal microscopy was used. Bacteria were stained with the red fluorescent nucleic acid stain SYTO59 according to the manufacturer's instructions. 100 μL of a 20 μM SYTO 59 solution was added to the biofilms and incubated at room temperature for 15 min. Free fluorophores were then removed by washing the biofilm with 100 μL physiological saline. CLSM was performed with a Nikon C1si confocal laser scanning microscope equipped with a Plan Apochromat 60 × 1.4 NA oil immersion objective lens (Nikon). The pictures were captured with NIS Elements Advanced Research package. SYTO59 was excited with a 636 nm diode laser (CVI Melles Griot, Albuquerque, NM, USA).

**AuNP penetration through biofilms**. The average hydrodynamic size and zeta potential of the AuNP were measured with DLS. After diluting the particles 1:100 in ultrapure water, 1 mL of the sample was transferred in a folded capillary cell and measured with the Zetasizer Nano-ZS (Malvern, Worcestershire, UK) in triplicate. The morphological structure of the AuNP was visualized with JEM 1400plus Transmission Electron Microscopy (TEM) (JEOL, Tokyo, Japan) operating at 60 kV. A 50 μL drop of a 1:10,000 dilution of the Au NP was blotted on formvar/C-coated hexagonal copper grids (EMS G200H-Cu) for 20 min and washed 5 times in ddH$_2$O. The extinction spectrum of the AuNP was recorded with a UV–VIS spectrophotometer (Nanodrop 2000, ThermoScientific, Wilmington, DE, USA).

After 24 h of growth, the supernatant was removed and 100 μL of an aqueous dispersion of AuNP (1.4E + 10 AuNP mL$^{-1}$) was added to the biofilm. To allow complete penetration, the biofilms were incubated for 15 min at room temperature. Next, the supernatant was removed and the biofilms were washed with 100 μL physiological saline in order to remove the excess AuNP.

To evaluate whether the AuNP were able to penetrate the biofilms, the location of AuNP inside the biofilms was investigated by CLSM. To visualize the biofilms, the bacteria were stained with SYTO59, as mentioned before and excited at 636 nm. AuNP were simultaneously visualized in reflection mode using the 561 nm laser line (CVI Melles Griot, Albuquerque, NM, USA).

**Formation and visualization of VNB inside biofilms**. A home-made optical set-up was used to generate and detect VNBs inside the biofilms, according to the optical design shown in Supplementary Figure 2[30]. The set-up is built around an inverted TE2000 epi-fluorescence microscope (Nikon, Nikon BeLux, Brussels, Belgium) equipped with a Plan Fluor 10× 0.3 NA objective lens (Nikon). An Optical Parametric Oscillator (OPO) laser (Opolette™ HE 355 LD, OPOTEK Inc., Faraday Ave, CA, USA) produces laser pulses of 7 ns tuned to 561 nm in order to excite the localized surface plasmon resonance of the AuNP while at the same time being compatible with optical filters in the set-up. The energy of each laser pulse is monitored with an energy meter (J-25MB-HE&LE, Energy Max-USB/RS sensors, Coherent) synchronized with the pulsed laser. In this study, a laser pulse fluence of 1.69 J cm$^{-2}$ was used. An automatic Prior Proscan III stage (Prior scientific Ltd., Cambridge, UK) is used to scan the sample through the 150 μm diameter laser beam (firing at 20 Hz) line by line with a scanning speed of 3 mm s$^{-1}$ and an interline distance of 0.15 mm (spatial overlap between subsequent laser pulses ensures that each location of the biofilm receives at least 1 laser pulse).

As VNBs efficiently scatter light, the generation of VNB inside biofilms could be detected by dark-field microscopy. Because of the short nature of VNB generation (lifetime < 1 μs), the camera (EMCCD camera, Cascade II: 512, Photometrics, Tucson, USA) was synchronized with the pulsed laser by an electronic pulse generator (BNC575, Berkeley Nucleonics Corporation, CA, USA). Dark field pictures were taken before illumination, during VNB formation and immediately after illumination, in order to elucidate conformational changes in the biofilm structure due to VNB formation. For dark field imaging, the biofilms were cultured 24 h at 37 °C in 50 mm glass bottom dishes (No. 1.5 coverslip) (MatTek Corporation, Ashland, USA).

**Combining VNB-induced cluster disruption and antibiotics**. The combined effect of VNB-mediated disturbance of the biofilm and antibiotic treatment was evaluated for 3 different organisms. The laser treatment described above was applied 1× or 10× as discussed in the text. After laser treatment, as described before, 100 μL supernatant was removed and the biofilms were treated with 100 μL tobramycin or control solution (0.9% NaCl (w/v)). In order to avoid photo-induced inactivation of the antibiotic, tobramycin was added after laser treatment. Tobramycin was dissolved in physiologic saline and sterilized by membrane filtration through a 0.22 μm filter. Tobramycin concentrations of 32, 16, and 1024 μg mL$^{-1}$ were used to treat *B. multivorans*, *P. aeruginosa*, and *S. aureus* biofilms, respectively. After treatment of 24 h at 37 °C, the sessile cells were washed with physiologic saline and harvested by 2 rounds of 5 min vortexing (900 rpm, Titramax 1000, Heidolph Instruments, Schwabach, Germany) and 5 min sonication (Branson 3510, Branson Ultrasonics Corp., Danbury, CT, USA). Next, the number of CFU/biofilm per condition was determined by plating ($n = 6 \times 3$).

**VNB-treatment enhances transport in biofilms**. In order to investigate our hypothesis that VNB can enhance antibiotic transport through biofilms, an in-depth microscopic analysis was performed of the penetration of a model fluorescent molecule into *P. aeruginosa* biofilms before and after VNB-treatment. Therefore, we compared the biofilm penetration of a model fluorophore (4 kDa FITC-dextrane) before and after VNB-formation by confocal microscopy. After cultivation of 24 h-old *P. aeruginosa* biofilms, 70 nm AuNP were added to the biofilms and the bacterial clusters were stained with 20 μM SYTO59. After a washing step with physiological saline, FITC-dextrane 4000 (Sigma) at a concentration of 4 mg mL$^{-1}$ was added to the biofilms. Biofilms were visualized in 3D by confocal microscopy. SYTO59 was excited with a 636 nm diode laser, while FITC-dextrane was excited using a 488 nm laser. In total, 20 different bacterial cell clusters were imaged in dual color ($n = 5 \times 4$). After laser treatment of the clusters, images were recorded at the exact same locations, by making use of reference points in the 96-well plate (diamond scratches).

The penetration of FITC-dextrane into the clusters was quantified off-line using ImageJ (National Institutes of Health). The image analysis protocol is displayed in Supplementary Figure 5. First, the image containing the signal of the bacteria was analyzed in order to define the bacterial cell clusters as regions of interest (ROI). After intensity thresholding, a binary image was created in which each pixel was designated as being either inside or outside the bacterial cluster. Only ROI with a surface area larger than 2 μm$^2$ were retained for analysis (avoiding inclusion of single bacterial cells). Finally, this set of ROIs were applied to the image containing the fluorescence of FITC-dextrane and the average fluorescence intensity of pixels within each ROI ($I_{in}$) was measured, as well as the fluorescence around the clusters ($I_{out}$). Autofluorescence was determined by recording images of SYTO59 stained biofilms without FITC-dextrane using the same image settings (performed in triplicate to determine the average value $\overline{I_{bg,in}}$ and $\overline{I_{bg,out}}$). Finally, FITC-dextrane penetration efficiency was calculated according to:

$$\frac{I_{in} - \overline{I_{bg,in}}}{I_{out} - \overline{I_{bg,out}}} \tag{1}$$

**Temperature assessment during VNB-treatment**. To confirm that the observed effects in bacterial killing are not caused by mere heat generation, the sample

temperature was monitored during laser illumination. A remote temperature sensor (Delta T Reference Thermistor, CHROMAPHOR, Germany) was inserted in the biofilm well to record the temperature during laser illumination (120 s for 1 well of a 96-well plate). Correct functioning of the temperature sensor was checked with ultrapure water of 4 and 50 °C.

**Bacterial dispersal during VNB-treatment**. Bacterial dispersal by VNB treatment was assessed by determining the number of colony forming units in the supernatants and biofilms of both *P. aeruginosa* and *S. aureus* via plate counting ($n = 3 \times 3$).

**In vivo toxicity assessment in *Caenorhabditis elegans***. To investigate the translational potential of VNB treatment, AuNP toxicity was evaluated in an in vivo toxicity test with *Caenorhabditis elegans* N2 (*glp-4; sek-1*). Both pristine and irradiated AuNP were included in the toxicity assay. In brief, *C. elegans* was maintained on NGM agar plates, which were seeded with *Escherichia coli* OP50. The plates were incubated for 6 h at 37 °C to allow the *E. coli* lawn to grow. Next, *C. elegans* worms were transferred from one NGM plate to another by the chunking technique. The chunked *C. elegans* plates were incubated for 2 weeks at 12 °C in order to cultivate the nematodes. Then, an egg-prep/bleaching step was performed in order to obtain a synchronized *C. elegans* population. Therefore, the nematodes were washed 2 times with physiological saline followed by a hypochlorite bleaching step. The obtained eggs were transferred to *E. coli* seeded NGM plates and incubated for 3 days at 25 °C. The synchronized nematodes (L4 stage) were then collected in OGM medium containing 95% M9 buffer, 5% Brain Heart Infusion Broth (Oxoid), and 10 μg mL$^{-1}$ cholesterol (Sigma-Aldrich). The nematode suspension was standardized, so that each well of a 96-well flat-bottomed microtiter plate contained 25 nematodes. *C. elegans* was fed with 25 μL of AuNP suspension (final concentration of $1.4E + 10$ AuNP mL$^{-1}$) for toxicity testing, whereas the controls were treated with OGM medium. The plates were incubated for 3 days at 25 °C and the number of living/dead nematodes were determined every 24 h by using an EVOS FL Auto Microscope (Life Technologies, USA) at 2× magnification ($n = 4 \times 6$).

**Statistical data analysis**. SPSS Statistics 24 (SPSS, Chicago, IL, USA) was used to analyze the data. The Shapiro–Wilk test was used to test the normality of the data sets. The one-way analysis of variance test and independent samples *T*-test were used for normal distributed data. The Kruskal–Wallis test and Mann–Whitney *U* test were used for non-normally distributed data. Differences with a *p*-value < 0.05 were considered significant.

## Data availability

The data that support the findings of the study are available in this article and its Supplementary Information files, or from the corresponding author upon request.

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

## Acknowledgements

This research was funded by Agency for Innovation by Science and Technology, University Ghent Special Research Fund (01G02215), and European Research Council (ERC) under the European Union's Horizon 2020 research and innovation program (Grant Agreement [648124]). R. Xiong is a postdoctoral fellow of the Research Foundation-Flanders (FWO-Vlaanderen) (1500418N).

## Author contributions

K. Braeckmans and T. Coenye developed the conceptual ideas, directed the project, and worked on the manuscript. S.C. De Smedt aided in interpreting the results and helped in writing the manuscript. E. Teirlinck carried out the experiments, analyzed the data, and wrote the manuscript with input from all authors. R. Xiong, T. Brans, K. Forier, J. Fraire, and H. Van Acker were involved in the experimental design, analysis, and writing of the manuscript. N. Matthijs assisted with the AuNP toxicity studies. R. De Rycke performed TEM imaging.

## Additional information

**Competing interests:** K. Braeckmans, T. Coenye, and S.C. De Smedt are listed as inventors in a patent application related to the procedure described in this article ("Disruption or alteration of microbiological films", WO 2017009039 A1, https://www.google.com/patents/WO2017009039A1?cl=en#npl-citations). The remaining authors declare no competing interests.

