## [Peer Review File · Nature Communications]

Reviewers' comments:

Reviewer #1 (Remarks to the Author):

The authors studied the use of gold nanoparticles with laser treatment to disrupt biofilms for better antibiotics penetration and efficacy. In this study, they were able to prove that gold nanoparticles penetrate the biofilms, and when treated with laser, forms nanobubbles. The formation VNB has already been previously shown, however, one of the major findings, that has not been seen before, is that these nanobubbles can be repeated with multiple laser treatment. The dark field microscopy images, as well as the videos show convincing data that these actually are occurring. All the efficacy studies were also done in a clear and convincing manner. As regards to nanoparticle synthesis, the authors did not present the step-by-step synthesis because the material was only bought from Nanopartz. It will be nice to have this but may not be feasible. However, the characterization clearly shows the NP properties that they need for the disruption of film using laser. These information, although was put in the supplementary figure, must also be written in the results section (absorption at ~560 nm, which is needed for laser treatment, etc). Overall, I think this paper is novel and is suitable for publication in Nature Communications.

Reviewer #2 (Remarks to the Author):

Excellent introductory work first to explore the use of gold nanoparticle assisted bubble disruption of bacterial biofilms and the synergy observed with sub inhibitory tobramycin in vitro. Exciting work that establishes the effect and benefit and reveals a lot of interesting followup research questions.

- How do the authors imagine that the nanoparticle dose ($1.4E+10$ AuNP/mL) and laser be applied in clinical practice? Additional details will help establish the impact and significance of the work.
- Figure 2 the authors should direct the reader to the location of the nanoparticles in the image - it is not clear.
- Quantification of the biofilm's elastic forces would be a nice addition to the manuscript.
- In Staph aureus it appears that the AuNPs used alone actually increase bacterial growth. Minor increases were also seen with other organisms. Then, when AuNPs were used with Tob alone some antagonism with Tob was seen with *B. multivorans*. Authors should consider how these observations might influence therapy if distribution of Tob or AuNPs within the biofilms or larger infection are not homogenous or exposed to laser treatments could this treatment be detrimental?
- the hypothesis that VNBs enhance drug penetration in biofilms should be tested with a diffusion assay or imaging of a model fluorescent drug if possible. The authors provide this hypothesis but do not address it directly - only providing indirect evidence.
- is penetration of the NP into the biofilm necessary for effect? One might imagine if the NPs bind to the biofilm that this could be used as a method of targeting to the biofilm, allowing multiple treatments.
- are the authors sure that individual NPs are responsible for observations rather than aggregated NPs?
- Future work should study the effect of this treatment on persisters.
- Figures 4 and 5: There is a confusion in the units used to represent biofilm survival. The results (y-axis) are %survival to untreated bacterial biofilm (log scale) but the caption says "data shown

are log CFU/biofilm". You can keep the data the way it is presented (and fix the caption) or change it to log CFU/biofilm (which would have larger values for control, up to 10^6 to 10^7). I would recommend presenting the results as the log of the survival CFU/biofilm or as log reduction as it is a more standard unit and can be useful to compare the results to other similar approaches reported in the literature.

- Line 571. Conclusion: The first sentence should be rephrased. The authors did not measure the penetration of antibiotics after the VNB pre-treatment. This is a proposed mechanism and was properly discussed but should not be stated in the conclusion as it was not tested.

Point-by-point responses to the referees' comments

Reviewer #1:

We would like to sincerely thank the reviewer for this positive evaluation of our manuscript.

1) *As regards to nanoparticle synthesis, the authors did not present the step-by-step synthesis because the material was only bought from Nanopartz. It will be nice to have this but may not be feasible.*

Following the reviewer's comment, we contacted the company providing the AuNP (Nanopartz) and obtained the following information regarding AuNP synthesis.

"Nanopartz 70 nm diameter gold nanoparticles are conjugated with myriad of different covalent SH-based branched and straight amine polymers with different MW from 5 kDa to 40 kDa, all manufactured by Nanopartz. Purification is a multistep dialysis resulting in less than 0.01% w-w reactants."

We added this information to the manuscript in the *Methods* section (*Materials and strains*, page 18).

"AuNP (Nanopartz, US) were diluted 1:60 in ultrapure water prior to use. According to the manufacturer's information, AuNP were conjugated with a myriad of different covalent SH-branched and straight amine polymers with different MW from 5 kDa to 40 kDa, all

manufactured by Nanopartz. For purification, a multistep dialysis was performed resulting in less than 0.01% w-w reactants.”

2) However, the characterization clearly shows the NP properties that they need for the disruption of film using laser. These information, although was put in the supplementary figure, must also be written in the results section (absorption at ~560 nm, which is needed for laser treatment, etc).

In light of the reviewer’s comment, we decided to move the data discussing AuNP characterization from *Supplementary Information* to the *Results* section in the main text (*AuNP penetration through biofilms*, page 6).

“Dynamic Light Scattering size data, zeta potential, TEM image and UV-VIS spectrum are displayed in Figure 2.”

Figure 2: **AuNP characterization.** Size distribution (a) and zeta potential (b) of AuNP, as determined by DLS. c) Morphological spherical structure of AuNP, visualized by TEM. Scale bar = 500 nm. d) UV-VIS spectrum of AuNP revealing a localized plasmon resonance peak at 549 nm.

Reviewer #2:

We would also like to sincerely thank reviewer 2 for the constructive comments that helped us further improving the quality of the manuscript.

3) *How do the authors imagine that the nanoparticle dose ($1.4E+10$ AuNP/mL) and laser be applied in clinical practice? Additional details will help establish the impact and significance of the work.*

It is a valid question which, however, needs further *in vivo* testing to find out what is the most optimal procedure. Nevertheless, at present we imagine the procedure to go as follows.

Considering the inherent limited penetration of light into tissue, the most likely applications are the treatment of topical infections, such as the treatment of wound infections. In this application the **AuNP** will likely be applied to the wounds as a topical suspension. After some incubation time this can be followed by a gentle rinsing step with a washing solution such as physiological saline to remove excess unbound AuNP. In the next step, the infected area will be irradiated with **pulsed laser** light to generate VNB. Due to the unsurpassed fine control of laser light we expect that unwanted side effects like tissue damage can be reduced to a minimum. While treatment of wounds with pulsed laser light is currently not performed to our knowledge, it is of note that pulsed lasers that are used for tattoo removal or port-wine stain treatment^{1,2} have the right characteristics for VNB generation, so that at least from a technology point the procedure should be clinically feasible. In the last step, the required **antimicrobial agent** will be administered to the VNB-treated wounds in the same way as is normally done in wound care.

Another potential application is the treatment of dental root canal infections, for which biofilm removal by laser-induced cavitation of the rinsing fluid is already being applied in the clinic³. Still, complete biofilm removal remains challenging and often incomplete due to the fact that fluid cavitation only affects the biofilm from the outside. Instead, if combined with AuNP to induce VNB from within the biofilms, it is expected that biofilm removal would become more effective. The AuNP can be dispersed in the disinfectant solution that is normally used during the laser procedure so that no extra steps will be needed. This should lead to more efficient

eradication of remaining biofilm cells not only because of the VNB shockwaves, but also due to a better penetration of the disinfectants that are present during the procedure.

In response to this comment, we added the following paragraphs to the *Discussion* section (page 13).

“Clearly, much work still needs to be done to further validate our new concept of VNB mediated biofilm disruption as well, including performing *in vivo* studies. Considering the inherent limited penetration of light into tissue, the most likely applications are the treatment of topical infections, such as the treatment of wound infections. In this application the AuNP will likely be applied to the wounds as a topical suspension. After some incubation time this can be followed by a gentle rinsing step with a washing solution such as physiological saline to remove excess unbound AuNP. In the next step, the infected area will be irradiated with pulsed laser light to generate VNB. Due to the unsurpassed fine control of laser light, we expect that unwanted side effects like tissue damage can be reduced to a minimum ~~due to the unsurpassed fine control of laser light~~. The use of light comes at its expense as well, which is that light penetration into tissues is fairly limited especially in the visible range. Although we used green-yellow light in this study (561 nm), it is important to note that this is not inherent to VNB generation, but only depends on the sensitizing particle being used. For instance, by switching e.g. to gold nanorods, the absorbance spectrum shifts to NIR wavelengths, allowing deeper penetration into tissues. ~~Still, we expect that applications will be mostly limited to more superficial infections, for example wound infections.~~ While treatment of wounds with pulsed laser light is currently not performed to our knowledge, it is of note that pulsed lasers that are used for tattoo removal or port-wine stain treatment^{1,2} have the right characteristics for VNB generation, so that at least from a technology point ~~of view clinical application of~~ the procedure should be clinically feasible. In the last step, the required antimicrobial agent will be administered to the VNB-treated wounds in the same way as is normally done in wound care. Another potential application is the treatment of dental root canal infections, for which biofilm removal by laser-induced cavitation of the rinsing fluid is already being applied in the clinic ³.

Still, complete biofilm removal remains challenging and often incomplete due to the fact that fluid cavitation only affects the biofilm from the outside. Instead, if combined with AuNP to induce VNB from within the biofilms, it is expected that biofilm removal would become more effective. The AuNP can be dispersed in the disinfectant solution that is normally used during the laser procedure so that no extra steps will be needed. This should lead to more efficient eradication of remaining biofilm cells not only because of VNB shockwaves, but also due to a better penetration of the disinfectants that are present during the procedure

4) Figure 2 the authors should direct the reader to the location of the nanoparticles in the image - it is not clear.

To avoid confusion, we changed the colours in Figure 3 '**AuNP penetration through biofilms**', so that bacteria are depicted in green and AuNP are coloured in magenta, as suggested in the *Nature Communications* guidelines. We also pointed out a few examples of AuNP with white arrow heads in all images to further guide the reader to the AuNP locations.

Figure 3: **AuNP penetration through biofilms**. Confocal images of AuNP (magenta) in biofilms of *B. multivorans*, *P. aeruginosa* and *S. aureus* (green). Left: Large-view 3-D confocal images showing various cell clusters. Right: Magnified view of the middle plane of selected cell clusters showing the presence of AuNP (some examples are indicated with white arrow heads) throughout the clusters. Scale bar = 50 μm .

5) Quantification of the biofilm's elastic forces would be a nice addition to the manuscript.

We appreciate the suggestion, although it is not easily addressed experimentally. Nevertheless, we tried to do so by performing rheology measurements on (treated) biofilms. Creep tests were performed using an AR1000 plate cone rheometer (TA Instruments). Creep tests have been reported for testing the viscoelastic behaviour of biofilms in which a constant stress is applied to the biofilm after which the resultant strain over time is measured⁴. We first compared the creep responses of 24h-old *P. aeruginosa* biofilms that were treated with AuNP (same conditions as in the proof-of-concept experiments) or CTRL (0.9% NaCl). A shear stress of 0,5 Pa was applied on the biofilms for 180 s and the resultant strain was recorded for another 180 s ($n = 3$). As can be seen in Figure X1, huge variations in the creep responses were observed in both CTRL as gold-treated samples. In the study of W. L. Jones *et al.* the viscoelastic properties of *P. aeruginosa* biofilms were also compared between different chemical treatments, and although they observed statistically significant alterations in the material properties of the biofilms, the experiment had to be repeated extensively ($n = 153$ for CTRL biofilms) in order to draw conclusions⁵. Based on this, although we gave it a sincere try, we hope the reviewer will appreciate that performing such extensive experiments really goes beyond the current scope, especially since it will not change anything on the outcome of the manuscript. But since we agree that it is of fundamental interest to better understand the underlying mechanisms, we added a comment in the *Discussion* (page 15) that future work should assess the viscoelastic change of biofilms after VNB-treatment. Here macroscopic rheology measurements could be combined with Single Particle Tracking microrheology experiments to come to an in-depth understanding of the visco-elastic changes by laser-induced VNB^{4,6}.

Figure X1: Creep responses of *P. aeruginosa* biofilms. Shear strain over time of *P. aeruginosa* biofilms treated with 0.9% NaCl (CTRL) or AuNP. The applied shear stress of 0.5 Pa was removed after 180 s.

6) *In Staph aureus it appears that the AuNPs used alone actually increase bacterial growth. Minor increases were also seen with other organisms. Then, when AuNPs were used with Tob alone some antagonism with Tob was seen with B. multivorans. Authors should consider how these observations might influence therapy if distribution of Tob or AuNPs within the biofilms or larger infection are not homogenous or exposed to laser treatments could this treatment be detrimental?*

The reviewer raises an interesting point, even though the effects are rather small and in most cases statistically not significant. In case of *S. aureus*, the amount of bacteria after AuNP incubation are indeed statistically higher than the CTRL. For both *B. multivorans* as *P. aeruginosa*, this was not the case. The differences in the amount of bacteria between tobramycin treated biofilms and tobramycin+AuNP treated biofilms are not statistically different for any of the tested organisms. Nevertheless, we agree with the reviewer that for the treatment's success, it will be crucial that the entire infected region is treated with AuNP and tobramycin and irradiated by pulsed laser light in order to reduce the infection. By scanning the laser beam in a (somewhat overlapping) raster pattern, a complete coverage of the infected area can be assured. In addition one could scan the wound multiple times (as it was observed that repeated VNB-formations inside biofilms is possible) in order minimize the chance that some small parts would be missed, e.g. due to slight movement by the patient during the scanning procedure.

In response to the reviewer's comment, we added the following information in the *Discussion* section on page 14.

"While treatment of wounds with pulsed laser light is currently not performed to our knowledge, it is of note that pulsed lasers that are used for tattoo removal or port-wine stain treatment^{1,2} have the right characteristics for VNB generation, so that at least from a technology point of view ~~clinical application of~~ the procedure should be clinically feasible. In the last step, the required **antimicrobial agent** will be administered to the VNB-treated wounds in the same

way as is normally done in wound care. It is of note that for the treatment's success, it will be crucial that the entire infected region is treated with AuNP and tobramycin and irradiated by pulsed laser light in order to reduce the infection. By scanning the laser beam in a (somewhat overlapping) raster pattern, a complete coverage of the infected area can be assured. In addition one could scan the wound multiple times (as it was observed that repeated VNB-formations inside biofilms is possible) in order minimize the chance that some small parts would be missed, e.g. due to slight movement by the patient during the scanning procedure. Another potential application is the treatment of dental root canal infections, for which biofilm removal by laser-induced cavitation of the rinsing fluid is already being applied in the clinic ³."

7) The hypothesis that VNBs enhance drug penetration in biofilms should be tested with a diffusion assay or imaging of a model fluorescent drug if possible. The authors provide this hypothesis but do not address it directly - only providing indirect evidence.

We would like to thank the reviewer for this comment. In reply to this remark, we performed an in-depth microscopic analysis of the penetration of a model fluorescent molecule into *P. aeruginosa* biofilms before and after VNB-treatment. Therefore we compared the biofilm penetration of a model fluorophore (4 kDa FITC-dextrane) before and after VNB-formation by confocal microscopy. After cultivation of 24h-old *P. aeruginosa* biofilms, 70 nm AuNP were added to the biofilms and the bacterial clusters were stained with 20 μ M SYTO59. After a washing step with physiological saline, FITC-dextrane 4000 (Sigma) at a concentration of 4 mg mL⁻¹ was added to the biofilms. Biofilms were visualized in 3D by confocal microscopy. SYTO59 was excited with a 636 nm diode laser, while FITC-dextrane was excited using a 488 nm laser. In total, 20 different bacterial cell clusters were imaged in dual colour ($n = 5 \times 4$). After laser treatment of the clusters, images were recorded at the exact same locations, by making use of reference points in the 96-well plate (diamond scratches). As displayed in Figure 7, after irradiating the biofilm with 1 laser pulse, the fluorescence of FITC-dextrane increased towards the center of the bacterial clusters, confirming the hypothesis that VNB mediated cluster disruption results in increased penetration of molecules into the clusters (see also Supplementary Movies). It was also observed that repeated VNB treatment resulted in a bigger flux of molecules towards the center of the clusters.

The penetration of FITC-dextrane into the clusters was quantified off-line using ImageJ (National Institutes of Health). The image analysis protocol is displayed in Supplementary Figure 5. First, the image containing the signal of the bacteria was analyzed in order to define the bacterial cell clusters as regions of interest (ROI). After intensity thresholding, a binary image was created in which each pixel was designated as being either inside or outside the bacterial cluster. Only ROI with a surface area larger than 2 μ m² were retained for analysis

(avoiding inclusion of single bacterial cells). Finally, this set of ROIs were applied to the image containing the fluorescence of FITC-dextrane and the average fluorescence intensity of pixels within each ROI (I_{in}) was measured, as well as the fluorescence around the clusters (I_{out}). Autofluorescence was determined by recording images of SYTO59 stained biofilms without FITC-dextrane using the same image settings (performed in triplicate to determine the average value $\overline{I_{bg,in}}$ and $\overline{I_{bg,out}}$). Finally, FITC-dextrane penetration efficiency was calculated according to:

$$\frac{I_{in} - \overline{I_{bg,in}}}{I_{out} - \overline{I_{bg,out}}} \quad (1)$$

As can be seen in the histograms in Figure 7, a single laser pulse increased FITC-dextrane penetration efficiency from 0.48 to 0.81 (and could be increased to 0.84 after 3 laser irradiations). In conclusion, the creation of vapour nanobubbles inside biofilms can locally disrupt the dense biofilm clusters and cause an increased influx of molecules into the clusters.

Figure 7: VNB-formation enhances the penetration of molecules into biofilm cell clusters. (a) FITC-dextrane 4 kDa penetration in an exemplary bacterial cell clusters of *P. aeruginosa* biofilms before and after VNB-treatment. Bacteria are displayed in green, while FITC-dextrane is depicted in magenta. (b) 3-D analysis of FITC-dextrane penetration into a total of 20 clusters ($n = 5 \times 4$) was quantified according to Eq. (1). The histograms show that FITC-dextrane penetration was markedly enhanced following 1x or 3x laser treatment.

Supplementary Figure 5: **Experimental set-up to evaluate the enhanced penetration through biofilms.** Bacteria are displayed in green, while FITC-dextrane is depicted in magenta.

We added this experiment into the 'Methods', 'Results' and 'Discussion' section in the main manuscript as follows:

1) *Methods* – '**VNB-treatment enhances transport in biofilms**' (page 21)

"In order to investigate our hypothesis that VNB can enhance antibiotic transport through biofilms, an in-depth microscopic analysis was performed of the penetration of a model fluorescent molecule into *P. aeruginosa* biofilms before and after VNB-treatment. Therefore we compared the biofilm penetration of a model fluorophore (4 kDa FITC-dextrane) before and after VNB-formation by confocal microscopy. After cultivation of 24h-old *P. aeruginosa* biofilms, 70 nm AuNP were added to the biofilms and the bacterial clusters were stained with 20 μ M SYTO59. After a washing step with physiological saline, FITC-dextrane 4000 (Sigma) at a concentration of 4 mg mL⁻¹ was added to the biofilms. Biofilms were visualized in 3D by confocal microscopy. SYTO59 was excited with a 636 nm diode laser, while FITC-dextrane was excited using a 488 nm laser. In total, 20 different bacterial cell clusters were imaged in dual colour ($n = 5 \times 4$). After laser treatment of the clusters, images were recorded at the exact same locations, by making use of reference points in the 96-well plate (diamond scratches).

The penetration of FITC-dextrane into the clusters was quantified off-line using ImageJ (National Institutes of Health). The image analysis protocol is displayed in Supplementary Figure 5. First, the image containing the signal of the bacteria was analyzed in order to define the bacterial cell clusters as regions of interest (ROI). After intensity thresholding, a binary image was created in which each pixel was designated as being either inside or outside the bacterial cluster. Only ROI with a surface area larger than 2 μ m² were retained for analysis (avoiding inclusion of single bacterial cells). Finally, this set of ROIs were applied to the image containing the fluorescence of FITC-dextrane and the average fluorescence intensity of pixels within each ROI (I_{in}) was measured, as well as the fluorescence around the clusters (I_{out}). Autofluorescence was determined by recording images of SYTO59 stained biofilms without FITC-dextrane using the same image settings (performed in triplicate to determine the average

value $\overline{I_{bg,in}}$ and $\overline{I_{bg,out}}$. Finally, FITC-dextrane penetration efficiency was calculated according to:

$$\frac{I_{in} - \overline{I_{bg,in}}}{I_{out} - \overline{I_{bg,out}}} \quad (1)$$

2) Results – **VNB-treatment enhances transport in biofilms** (page 9)

“As displayed in Figure 7, after irradiating the biofilm with 1 laser pulse, the fluorescence of FITC-dextrane increased towards the center of the bacterial clusters, confirming the hypothesis that VNB mediated cluster disruption results in increased penetration of molecules into the clusters (see also Supplementary Movies). It was also observed that repeated VNB treatment resulted in a bigger flux of molecules towards the center of the clusters.

As can be seen in the histograms in Figure 7, a single laser pulse increased FITC-dextrane penetration efficiency from 0.48 to 0.81 (and could be increased to 0.84 after 3 laser irradiations). In conclusion, the creation of vapour nanobubbles inside biofilms can locally disrupt the dense biofilm clusters and cause an increased influx of molecules into the clusters.”

3) Discussion (page 15)

“Indeed, the use of VNB significantly increased the effect of tobramycin by ~80 times, ~20 times and ~25 times in case of *B. multivorans*, *P. aeruginosa* and *S. aureus* biofilms, respectively. This is in line with our hypothesis that VNB treatment increases the drug flux towards deeper cell layers by creating more space between the cells. Indeed, it was observed that VNB cause an influx of molecules into the clusters after laser irradiation. Repeated VNB treatment resulted in an even bigger flux of molecules towards the center of the clusters.”

8) Is penetration of the NP into the biofilm necessary for effect? One might imagine if the NPs bind to the biofilm that this could be used as a method of targeting to the biofilm, allowing multiple treatments.

We thank the reviewer for this comment. We believe that efficient penetration of AuNP towards the center of biofilm clusters is a keystone of the treatment's success, because in this way the vapour nanobubbles are created inside the clusters, enabling a disruption force from within – instead of only disrupting the biofilm from the outer edges. Nevertheless, we fully agree with the reviewer that for the eventual *in vivo* applications, it could be beneficial to functionalize AuNP with ligands in order to target bacterial biofilms and minimize off-targets effects on the surrounding tissue.

In response to the reviewers' comment, we added the following sentence to the *Discussion* section on page 12.

“Efficient penetration of AuNP towards the center of biofilm clusters is believed to be a keystone of the treatment's success, because in this way the vapour nanobubbles are created inside the clusters, enabling a disruption force from within – instead of only disrupting the biofilm from the outer edges.”

9) Are the authors sure that individual NPs are responsible for observations rather than aggregated NPs?

The confocal images have shown that not only single AuNP but also AuNP aggregates are present inside the biofilm. Both can cause VNB-induced cluster disruption (although in a different extent). This is addressed in the manuscript on line 218-222, page 12.

10) Future work should study the effect of this treatment on persisters.

We agree that this is an interesting point for further exploration in the future. This dormant, multidrug-tolerant subpopulation of biofilm cells have been proposed as an important factor in biofilm resilience⁷. It would be interesting to see whether the increased transport of antibiotics by VNB-mediated disruption of biofilms would affect the number of persisters present in the biofilm population and their ability to sustain antibiotic killing.

We added the following paragraph in the *Discussion* section, page 15.

“Besides increasing net mass transport, disruption of biofilms could also have an impact on other tolerance mechanisms that are derived from their complex three-dimensional structure⁸. In future work, the effect of VNB-disruption on persisters will be studied. This dormant, multidrug-tolerant subpopulation of biofilm cells have been proposed as an important factor in biofilm resilience⁷. It would be interesting to see whether the increased transport of antibiotics by VNB-mediated disruption of biofilms would affect the number of persisters present in the biofilm population and their ability to sustain antibiotic killing.”

11) Figures 4 and 5: There is a confusion in the units used to represent biofilm survival. The results (y-axis) are %survival to untreated bacterial biofilm (log scale) but the caption says "data shown are log CFU/biofilm". You can keep the data the way it is presented (and fix the caption) or change it to log CFU/biofilm (which would have larger values for control, up to 10^6 to 10^7). I would recommend presenting the results as the log of the survival CFU/biofilm or as log reduction as it is a more standard unit and can be useful to compare the results to other similar approaches reported in the literature.

We understand the confusion of the reviewer and therefore changed the captions of Figure 5 and 6 in the manuscript (page 31).

Figure 5: Anti-biofilm effect of combined treatment of 1x VNB formation and tobramycin. Data shown are log of the % survival CFU/biofilm compared to untreated control. ~~Log CFU/biofilm~~ (average \pm SEM) in (a) *B. multivorans*, (b) *P. aeruginosa*, and (c) *S. aureus* biofilms.

Figure 6: Biofilm survival after 10x VNB formation followed by tobramycin treatment. Log of the % survival CFU/biofilm compared to untreated control ~~Log CFU/biofilm~~ (average \pm SEM) in (a) *P. aeruginosa* and (b) *S. aureus* biofilms.

12) Line 571. Conclusion: The first sentence should be rephrased. The authors did not measure the penetration of antibiotics after the VNB pre-treatment. This is a proposed mechanism and was properly discussed but should not be stated in the conclusion as it was not tested.

We take the point that in the previous version of the manuscript this conclusion could not be drawn as it was not explicitly tested. However, as discussed in Comment 7, the new experimental data clearly confirm that VNB-treatment causes a strongly enhanced influx of molecules into the biofilm clusters. With these additional data we believe the conclusion is valid.

References:

1. Husain, Z. & Alster, T. S. The role of lasers and intense pulsed light technology in dermatology. *Clin. Cosmet. Investig. Dermatol.* **9**, 29 (2016).
2. Gianfaldoni, S. *et al.* An Overview of Laser in Dermatology: The Past, the Present and ... the Future (?). *Open Access Maced. J. Med. Sci.* **5**, 526 (2017).
3. Meire, M. A. Application of lasers in cleaning, disinfection and sealing of root canals: an in vitro study. (Ghent University, 2011).
4. Billings, N., Birjiniuk, A., Samad, T. S., Doyle, P. S. & Ribbeck, K. Material properties of biofilms—a review of methods for understanding permeability and mechanics. *Reports Prog. Phys.* **78**, 36601 (2015).
5. Jones, W. L., Sutton, M. P., McKittrick, L. & Stewart, P. S. Chemical and antimicrobial treatments change the viscoelastic properties of bacterial biofilms. *Biofouling* **27**, 207–215 (2011).
6. Peterson, B. W. *et al.* Viscoelasticity of biofilms and their recalcitrance to mechanical and chemical challenges. *FEMS Microbiol. Rev.* **39**, 234–245 (2015).
7. Lewis, K. Multidrug Tolerance of Biofilms and Persister Cells. in 107–131 (Springer, Berlin, Heidelberg, 2008). doi:10.1007/978-3-540-75418-3_6
8. Flemming, H.-C. *et al.* Biofilms: an emergent form of bacterial life. *Nat. Rev. Microbiol.* **14**, 563–575 (2016).

REVIEWERS' COMMENTS:

Reviewer #3 (Remarks to the Author):

The manuscript by Teirlinck et al. presents an elegant and novel approach to treat biofilms, based on vapour nanobubbles generated by AuNP exposed to a laser beam. In general I think that the manuscript is scientifically sound and relevant for the biofilm community and the AMR field.

I just have a few comments:

While the video supplementary files support the author's findings that AuNP enter the biofilm, Figure 3 selection of images is not totally convincing. Perhaps the less clear is the one for *B. multivorans*, where all the AuNP seem to be on the periphery of the biofilm. Some of the CLSM images seem to lack a scale bar, particularly in the lower magnification images (and Figure S1, for instance)

Lines 146-147: How did the authors guarantee that each location of the sample in the 96-well plates received only 1 laser pulse? The laser beam irradiates in a circular shape, and this to me means that either some overlapping of the laser occurred or there were sections of the well that were not irradiated.

Line 155 and others in this section: The method used (plating in culture medium) assesses a decrease in cultivable, rather than viable cells. Are these terms interchangeable?

I couldn't find the amount of biofilm (in CFU's per cm²) before the treatment for any of the monospecies biofilm. This information is valuable for comparison purposes and to better understand the numbers of CFU's that are present after treatment – we only have % survival.

Section starting in line 182: While the added experiment is noteworthy, a little more detail here could be useful for readers. Without referring to the materials and methods I could not understand if the analysis was performed at a single time point or continuously, and what was meant by "bigger flux of molecules towards the center of the clusters" as opposed to "increased penetration of molecules into the clusters" in the previous sentence. After how long were the pictures taken?

Nuno Azevedo

Laser-Induced Vapour Nanobubbles Improve Drug Diffusion and Efficiency in Bacterial Biofilms

Point-by-point responses to the referees' comments

Reviewer #3 (Remarks to the Author):

- 5 The manuscript by Teirlinck et al. presents an elegant and novel approach to treat biofilms, based on vapour nanobubbles generated by AuNP exposed to a laser beam. In general I think that the manuscript is scientifically sound and relevant for the biofilm community and the AMR field.

I just have a few comments:

- 10 **1) While the video supplementary files support the author's findings that AuNP enter the biofilm, Figure 3 selection of images is not totally convincing. Perhaps the less clear is the one for *B. multivorans*, where all the AuNP seem to be on the periphery of the biofilm.**

- 15 We thank the reviewer for pointing this out and inserted another image that shows more clearly that AuNP were present inside the cell clusters – and not only on the outside. We also tried to further enhance the contrast of all figures so that the small AuNP inside the cell clusters can be seen more clearly.

20 **Figure 3: AuNP penetration through biofilms.** Confocal images of AuNP (magenta) in biofilms of *B. multivorans*, *P. aeruginosa* and *S. aureus* (green). Left: Large-view 3-D confocal images showing various cell clusters. Width = 212 μm , Height = 212 μm , Depth = 35 μm (*B. multivorans*, *P. aeruginosa*), 30 μm (*S. aureus*). Right: Magnified view of the middle plane of selected cell clusters showing the presence of AuNP (some examples are indicated with white arrow heads) throughout the clusters. Scale bar = 50 μm .

2) **Some of the CLSM images seem to lack a scale bar, particularly in the lower magnification images (and Figure S1, for instance)**

30 In light of this comment, we added the following information to the Figure legends on page 30-31 and in the Supplementary Information.

Figure 3: AuNP penetration through biofilms. Confocal images of AuNP (magenta) in biofilms of *B. multivorans*, *P. aeruginosa* and *S. aureus* (green). Left: Large-view 3-D confocal images showing various cell clusters. Width = 212 μm , Height = 212 μm , Depth = 35 μm (*B. multivorans*, *P. aeruginosa*), 30 μm (*S. aureus*). Right: Magnified view of the middle plane of selected cell clusters showing the presence of AuNP (some examples are indicated with white arrow heads) throughout the clusters. Scale bar = 50 μm .

Figure 7: VNB-formation enhances the penetration of molecules into biofilms. (a) FITC-dextrane 4 kDa penetration in an exemplary bacterial cell clusters of *P. aeruginosa* biofilms before and after VNB-treatment. Bacteria are displayed in green, while FITC-dextrane is depicted in magenta. Width = 212 μm , Height = 212 μm , Depth = 18 μm (b) 3-D analysis of FITC-dextrane penetration into a total of 20 clusters ($n = 5 \times 4$) was quantified according to Eq. (1). The histograms show that FITC-dextrane penetration was markedly enhanced following 1x or 3x laser treatment.

Supplementary Figure 1:

45 **Evaluation of the biofilm structure by CLSM.** Confocal image of 24-hours old biofilms of (left) *B. multivorans*, (center) *P. aeruginosa* and (right) *S. aureus*, grown in Mueller Hinton Broth, Lysogeny Broth and simulated wound fluid, respectively. Bacteria were stained with 20 μM SYTO59. Scale bar = 50 μm .

50 **3) Lines 146-147: How did the authors guarantee that each location of the sample in the 96-well plates received only 1 laser pulse? The laser beam irradiates in a circular shape, and this to me means that either some overlapping of the laser occurred or there were sections of the well that were not irradiated.**

55 The reviewer is correct in this regard. The illumination area is circular, which means there is spatial overlap between subsequent laser pulses in order to make sure that each location of the biofilm receives at least 1 laser pulse. To be precise, about 50% of the sample receives a single laser pulse, while 50% in the overlapping areas receives two pulses. We added this information to the text on page 20.

60 “An automatic Prior Proscan III stage (Prior scientific Ltd., Cambridge, UK) is used to scan the sample through the 150 μm diameter laser beam (firing at 20 Hz) line by line with a scanning speed of 3 mm s^{-1} and an interline distance of 0.15 mm (spatial overlap between subsequent laser pulses ensures that each location of the biofilm receives at least 1 laser pulse).”

65 **4) Line 155 and others in this section: The method used (plating in culture medium) assesses a decrease in cultivable, rather than viable cells. Are these terms interchangeable?**

70 It is a matter of definition but one can argue that “viable” and “cultivable” bacteria are not interchangeable. Although plate counting is the standard method to determine bacterial counts, it’s most important drawback is that it is based on the enumeration of cultivable bacteria on suitable media, and thus doesn’t take into account the viable but non-culturable (VBNC) bacteria. In order to avoid confusion, we therefore changed lines 117-119 and 121-122 as follows:

75 “Treatment with AuNP or pulsed laser irradiation alone did not have a significant effect on the amount of CFUs in any of the biofilms ($p > 0.05$).”

“Also VNB treatment by itself did not significantly decrease the number of CFUs.”

80 **5) I couldn’t find the amount of biofilm (in CFU’s per cm²) before the treatment for any of the monospecies biofilm. This information is valuable for comparison purposes and to better understand the numbers of CFU’s that are present after treatment – we only have % survival.**

We agree with the reviewer that this information could be helpful for comparison purposes, and therefore added the following lines in the Figure legends of Figure 5 and 6 on page 30-31:

85 **Figure 5: Anti-biofilm effect of combined treatment of 1x VNB formation and tobramycin.** Data shown are log of the % survival CFU/biofilm compared to the untreated control (average \pm SEM) in (a) *B. multivorans*, (b) *P. aeruginosa*, and (c) *S. aureus* biofilms. Control = 0.9% NaCl solution, AuNP = only addition of 70 nm spherical gold nanoparticles, Laser = only laser treatment (each location of the biofilm received 1 laser pulse), VNB = addition of AuNP with subsequent laser treatment created vapour nanobubbles, Tob = tobramycin treatment for 24 hours at 37°C, AuNP + Tob = addition of AuNP with subsequent tobramycin treatment, Laser + Tob = laser and subsequent tobramycin treatment, VNB + Tob = addition of AuNP with subsequent laser and tobramycin treatment. CTRL contained $9 \times 10^6 \pm 3 \times 10^6$ CFU/biofilm, $2 \times 10^7 \pm 1 \times 10^7$ CFU/biofilm or $2 \times 10^6 \pm 2 \times 10^6$ CFU/biofilm for *B. multivorans*, *P. aeruginosa* or *S. aureus* biofilms respectively. Each antibiofilm effect was tested in 6 biological repeats and each biological repeat consisted of 3 technical repeats ($n = 6 \times 3$) ($p < 0,05$; The Shapiro-Wilk test was used to test the normality of the data sets. The one-way analysis of variance test and independent samples T-test were used for normal distributed data. The Kruskal-Wallis test and ManWhitney U test were used for non-normally distributed data).

90

95

Figure 6: Biofilm survival after 10x VNB formation followed by tobramycin treatment. Log of the % survival CFU/biofilm compared to the untreated control (average \pm SEM) in (a) *P. aeruginosa* and (b) *S. aureus* biofilms. Control = 0.9% NaCl solution, AuNP = only addition of 70 nm spherical gold nanoparticles, Laser = only laser treatment (each location of the biofilm received 10 laser pulses), VNB = addition of AuNP with subsequent laser treatment to form vapour nanobubbles, Tob = tobramycin treatment for 24 hours at 37°C, AuNP + Tob = addition of AuNP with subsequent tobramycin treatment, Laser + Tob = laser and subsequent tobramycin treatment, VNB + Tob = addition of AuNP with subsequent laser and tobramycin treatment. CTRL contained $2 \times 10^7 \pm 7 \times 10^6$ CFU/biofilm or $1 \times 10^6 \pm 3 \times 10^5$ CFU/biofilm for *P. aeruginosa* or *S. aureus* biofilms respectively. Each antibiofilm effect was tested in 4 biological repeats and each biological repeat consisted of 3 technical repeats ($n = 4 \times 3$) ($p < 0,05$; The Shapiro-Wilk test was used to test the normality of the data sets. The one-way analysis of variance test and independent samples T-test were used for normal distributed data. The Kruskal-Wallis test and ManWhitney U test were used for non-normally distributed data).

6) Section starting in line 182: While the added experiment is noteworthy, a little more detail here could be useful for readers. Without referring to the materials and methods I could not understand if the analysis was performed at a single time point or continuously, and what was meant by “bigger flux of molecules towards the center of the clusters” as opposed to “increased penetration of molecules into the clusters” in the previous sentence. After how long were the pictures taken?

The confocal pictures (z-stacks) were taken before and right after VNB formation. The analysis via ImageJ was done on every image of the z-stacks before laser illumination and the corresponding images after laser illumination.

To make this part more clear, we adjusted the text on page 9 as follows:

In order to investigate our hypothesis that VNB can enhance antibiotic transport through biofilms, the biofilm penetration of FITC-dextrane was compared before and after VNB-formation by confocal microscopy. As displayed in Figure 7, after irradiating the biofilm with 1 laser pulse, the fluorescence of FITC-dextrane increased towards the center of the bacterial clusters, confirming the hypothesis that VNB mediated cluster disruption results in increased penetration of molecules into the clusters (see also Supplementary Movies 6-11). It was also observed that repeated VNB treatment could further increase the amount of fluorescence of FITC-dextrane that penetrated in the bacterial clusters.

The penetration of FITC-dextrane into the clusters was quantified off-line using ImageJ (National Institutes of Health). As displayed in Supplementary Figure 5, FITC-dextrane penetration efficiency was determined by comparing FITC-dextrane fluorescence intensity inside versus around the bacterial clusters (after adjusting for the autofluorescence). Image analysis revealed that a single laser pulse increased the FITC-dextrane penetration

efficiency from 0.48 to 0.81 (and could be increased to 0.84 after 3 laser irradiations). In conclusion, the creation of vapour nanobubbles inside biofilms can locally disrupt the dense biofilm clusters and cause an increased penetration of molecules into the clusters.”